# Seismic arrival-time picking on distributed acoustic sensing data using semi-supervised learning

Weiqiang Zhu [1,2] ✉, Ettore Biondi [1], Jiaxuan Li [1], Jiuxun Yin[1], Zachary E. Ross [1] & Zhongwen Zhan[1]

Distributed Acoustic Sensing (DAS) is an emerging technology for earthquake monitoring and subsurface imaging. However, its distinct characteristics, such as unknown ground coupling and high noise level, pose challenges to signal processing. Existing machine learning models optimized for conventional seismic data struggle with DAS data due to its ultra-dense spatial sampling and limited manual labels. We introduce a semi-supervised learning approach to address the phase-picking task of DAS data. We use the pre-trained PhaseNet model to generate noisy labels of P/S arrivals in DAS data and apply the Gaussian mixture model phase association (GaMMA) method to refine these noisy labels and build training datasets. We develop PhaseNet-DAS, a deep learning model designed to process 2D spatio-temporal DAS data to achieve accurate phase picking and efficient earthquake detection. Our study demonstrates a method to develop deep learning models for DAS data, unlocking the potential of integrating DAS in enhancing earthquake monitoring.

Distributed acoustic sensing (DAS) is a rapidly developing technology that can turn a fiber-optic cable of up to one hundred kilometers into an ultra-dense array of seismic sensors spaced only a few meters apart. DAS uses an interrogator unit to send laser pulses into an optical fiber and measure the Rayleigh back-scattering from the internal natural flaws of the optical fiber. By measuring the tiny phase changes between repeated pulses, DAS can infer the longitudinal strain or strain rate over time along a fiber-optic cable[1–3]. Previous studies have demonstrated that DAS can effectively record seismic waves[4–9]. Compared with traditional forms of seismic acquisition, DAS has several potential advantages in earthquake monitoring. It provides unprecedented channel spacing of meters compared with tens-of-kilometers spacing of seismic networks. DAS can also take advantage of dark fibers (i.e., unused strands of telecommunication fiber) at a potentially low cost. Furthermore, DAS is suitable for deployment and maintenance in challenging environments, such as boreholes, offshore locations, and glaciers. New DAS interrogator units are becoming capable of longer

sensing ranges at a lower cost with the development of high-speed Internet infrastructure[1]. Thus, DAS is a promising technology for improved earthquake monitoring and is under active research. However, applying DAS to routine earthquake monitoring tasks remains challenging due to the lack of effective algorithms for detecting earthquakes and picking phase arrivals, coupled with the high data volume generated by thousands of channels. The ultra-high spatial resolution of fiber-optic sensing is a significant advantage compared to seismic networks but also presents a challenge for traditional data processing algorithms designed for single- or three-component seismometers. For example, the commonly used STA/LTA (short-term averaging over long-term averaging) method[10] is ineffective for DAS because DAS recordings are much noisier than dedicated seismometer data due to factors such as cable-ground coupling and sensitivity to anthropogenic noise. STA/LTA operates on a single DAS trace and therefore does not effectively utilize the dense spatial sampling provided by DAS. Template matching is another effective earthquake

---

[1]Seismological Laboratory, Division of Geological and Planetary Sciences, California Institute of Technology, Pasadena, CA, USA. [2]Berkeley Seismological Laboratory, Department of Earth and Planetary Science, University of California, Berkeley, CA, USA. ✉e-mail: zhuwq@berkeley.edu

detection method, particularly for detecting tiny earthquake signals[11–14]. However, the requirement of existing templates and high computational demands limit its applicability for routine earthquake monitoring[15].

Deep learning, especially deep neural networks, is currently the state-of-the-art machine learning algorithm for many tasks, such as image classification, object detection, speech recognition, machine translation, text/image generation, and medical image segmentation[16]. Deep learning is also widely used in earthquake detection[17–22] for studying dense earthquake sequences[23–28] and routine monitoring seismicity[29–33]. Compared to the STA/LTA method, deep learning is more sensitive to weak signals of small earthquakes and more robust to noisy spikes that cause false positives for STA/LTA. Compared to the template matching method, deep learning generalizes similarity-based search without requiring precise seismic templates and is significantly faster. Neural network models automatically learn to extract common features of earthquake signals from large training datasets and are able to generalize to earthquakes outside the training samples. For example, the PhaseNet model, which is a deep neural network model trained using earthquakes in Northern California, performs well when applied to tectonic[24,25], induced[23,26], and volcanic earthquakes[34,35] globally.

One critical factor in the success of deep learning in earthquake detection and phase picking is the availability of many phase arrival-time measurements manually labeled by human analysts over the past few decades. For example Ross et al.[18] collected ~1.5 million pairs of P and S picks from the Southern California Seismic Network. Zhu and Beroza[19] employed ~700k P and S picks from the Northern California Seismic Network. Michelini et al.[36] built a benchmark dataset of ~1.2 million seismic waveforms from the Italian National Seismic Network. Zhao et al.[37] formed a benchmark dataset of ~2.3 million seismic waveforms from the China Earthquake Networks. Mousavi et al.[38] created a global benchmark dataset (STEAD) of ~1.2 million seismic waveforms; Several other benchmark datasets are also developed for training deep learning models[39,40]. Although many DAS datasets have been collected[41] and more continue to be collected, most of these datasets have not yet been analyzed by human analysts. Manually labeling a large DAS dataset can be costly and time-consuming. As a result, there are limited applications of deeplearning for DAS data. Most works focus on earthquake detection using a small dataset[42–44]. Accurately picking phase arrivals is an unsolved challenge for DAS data, hindering its applications to earthquake monitoring.

There have been a number of approaches proposed to train deep learning models with little or no manual labeling, such as data augmentation[45], simulating synthetic data[46–48], fine-tuning and transfer learning[49,50], self-supervised learning[51], and unsupervised learning[52,53]. However, those methods have not proven effective in picking phase arrival time on DAS data. One challenge is the difference in the mathematical structures between seismic data and DAS data, i.e., ultra-dense DAS arrays and sparse seismic networks, which complicate implementation of model fine-tuning or transfer learning. Additionally, phase arrival-time picking requires high temporal accuracy, which is difficult to achieve through self-supervised or unsupervised learning without accurate manual picks. Semi-supervised learning provides an alternative approach, which is designed for problems with limited labeled data and abundant unlabeled data[54,55]. There are several ways to utilize a large amount of unlabeled data as weak supervision to improve model training. One example is the Noisy Student method[54], which consists of three main steps: (1) training a teacher model on labeled samples, (2) using the teacher to generate pseudo labels on unlabeled samples, and (3) training a student model on the combination of labeled and pseudolabeled data. Thus, the Noisy Student method can leverage a substantial amount of unlabeled data to improve model accuracy and robustness.

In this work, we present a semi-supervised learning approach for training a deep learning model to pick seismic phase arrivals in DAS data without needing manual labels. Despite the differences in data modalities between DAS data (i.e., spatio-temporal) and seismic data (i.e., time series), the recorded seismic waveforms exhibit similar characteristics. Based on this connection, we investigate using semi-supervised learning to transfer the knowledge learned by PhaseNet for picking P and S phase arrivals from seismic data to DAS data. We develop a new neural network model, PhaseNet-DAS, that utilizes spatial and temporal information to consistently pick seismic phase arrivals across hundreds of DAS channels. We borrow a similar idea of pseudo labeling[56] to generate pseudo labels of P and S arrival picks in DAS data in order to train deep learning models using unlabeled DAS data. We extend the semi-supervised learning method to bridge two data modalities of 1D seismic waveforms and 2D DAS recordings so that we can combine the advantages of the abundant manual labels of seismic data and the large volume of DAS data. We demonstrate the semi-supervised learning approach by training two models. The PhaseNet-DAS v1 is trained using pseudo labels generated by PhaseNet to transfer phase picking capability from seismic data to DAS data. The PhaseNet-DAS v2 is trained using pseudo labels generated by PhaseNet-DAS v1 to further improve model performance similar to the Noisy Student method. Unless specified otherwise, we default to using the PhaseNet-DAS v2 model for evaluation in the following sections. We test our method using DAS arrays in Long Valley and Ridgecrest, CA, and evaluate the performance of PhaseNet-DAS in terms of number of phase picks, phase association rate, phase arrival time resolution, and earthquake detection and location.

## Results
### Phase picking performance
One challenge in picking phase arrivals in DAS data is the presence of strong background noise, as fiber-optic cables are often installed along roads or in urban environments and DAS is highly sensitive to surface waves. The waveforms of traffic signals have certain resemblance to earthquake signals with sharp emergence of first arrivals and strong surface waves, which leads to many false detections by the pre-trained PhaseNet model. Traffic signals are usually locally visible over short distances of a few kilometers without clear body waves. In contrast, earthquake signals tend to be much stronger and recorded by an entire DAS array with both body and surface waves present. PhaseNet-DAS uses both spatial and temporal information across multiple channels of a DAS array, making it more robust to traffic noise. Figure 1 shows four examples of earthquake signals that can be observed in sections of the DAS array. Due to strong background noise, we can see that PhaseNet detects many false P and S arrivals. However, PhaseNet-DAS predictions have fewer false detections and are consistent across channels with reduced variation in the picked arrival times. We apply both models to all events of four DAS cables and compare the number of associated picks, since picks that can be successfully associated are more indicative of true positives. After applying the phase associator GaMMA[57], the rates of associated phase picks increase from 59% - 69% for PhaseNet to 89% - 92% for PhaseNet-DAS (Table S1).

In addition to traffic noise, other factors such as poor ground coupling and instrumental noise make the signal-noise ratio (SNR) of DAS data generally lower than that of seismic data. The low SNR makes it challenging to detect and pick phase arrivals on DAS data. The PhaseNet model pre-trained on seismic data can detect high SNR events, but struggles with low SNR events in DAS data (Fig. 2). After re-training using semi-supervised learning on DAS data, the PhaseNet-DAS model significantly improves detections of low SNR events. PhaseNet-DAS v1 detects 2–5 times more events than PhaseNet across four DAS cables, and PhaseNet-DAS v2 enhances detection sensitivity by an additional 25–50% compared to PhaseNet-DAS v1 (Fig. 2). Moreover, the number of phase picks per events also significantly increases for both high and low SNR events after re-training (Fig. S1). This demonstrates that the PhaseNet-DAS model, which is designed to

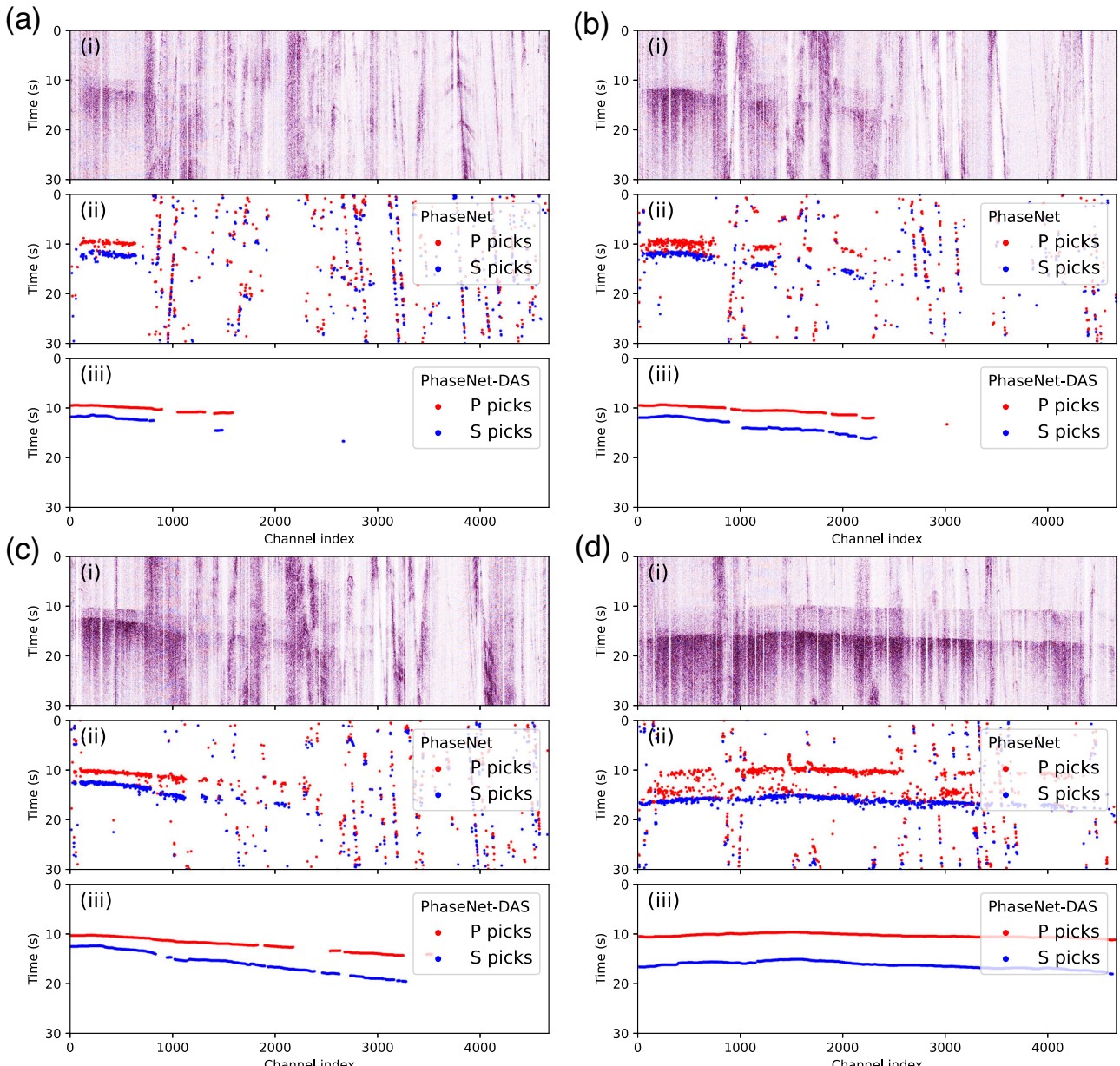

**Fig. 1 | Examples of noisy picks predicted by PhaseNet and improved picks predicted by PhaseNet-DAS. a–d** Four examples with different signal-to-noise ratios. Each sub-panel shows (i) DAS recordings of 30 s and 5000 channels, (ii) the PhaseNet picks, and (iii) the PhaseNet-DAS picks.

use coherent spatial information, can effectively detect weaker earthquake signals recorded by DAS and pick P and S picks on more DAS channels than the PhaseNet model, which is designed for 3-component seismic waveforms.

The noisy condition of DAS recording could also impact the temporal precision of picked phase arrival-times for both manual labeling and automatic algorithms. Because we lack manual labels of P and S arrivals as benchmarks, we evaluate the temporal accuracy of PhaseNet-DAS's picks indirectly. First, we compared the automatically picked phase arrival-times with the theoretical phase arrival-times using a 1D velocity model[58]. For events within ~100 km, the automatic picks have small time residuals within 2 s, while the time residuals increase with epicenter distances (Fig. S2). This discrepancy arises not from imprecise automatic picks, but from differences between the true 3D velocity model and the 1D velocity model we used. Then, we conducted a more precise analysis of the automatically picked phase arrival-times by comparing the differential arrival-times between two

events measured using waveform cross-correlation. Waveform cross-correlation is commonly used for earthquake detection (known as template matching or match filtering)[11–14], measuring differential travel-time[59–62] and relative polarity[63]. Cross-correlation achieves a high temporal resolution of the waveform sampling rate or super-resolution using interpolation techniques. We cut a 4-s time window around the arrival picked by PhaseNet-DAS, applied a band-pass filter between 1 Hz and 10 Hz, and calculated the cross-correlation between event pairs. The differential time was determined from the peak of the cross-correlation profile. Because DAS waveforms are usually much noisier than seismic waveforms and have low cross-correlation coefficients, we further improved the robustness of differential time measurements using multi-channel cross-correlation[64,65] to accurately extract the peaks across multiple cross-correlation profiles. We selected 2539 event pairs and ~9 millions differential time measurements for both P and S waves as the reference to evaluate the temporal accuracy of PhaseNet-DAS picks. Figure 3 shows the statistics of these two

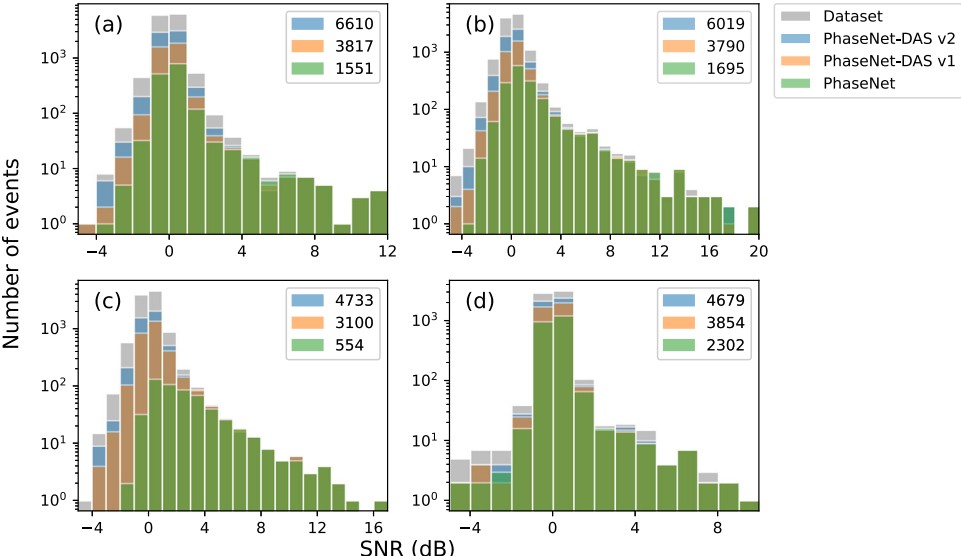

**Fig. 2 | SNR distributions of detected events across four DAS arrays. a** Mammoth north, **b** Mammoth south, **c** Ridgecrest north, **d** Ridgecrest south. The locations of the four DAS array are shown in Fig. 8. The SNR is calculated using two 5-s windows before and after the theoretical P wave arrival time. The PhaseNet-DAS v1 and v2 models are from the first and second iterations of the semi-supervised learning procedures illustrated in Fig. 6.

differential time measurements. If we assume the differential time measurements by waveform cross-correlation are the ground truth, the errors of differential time measurements by PhaseNet-DAS have a mean of 0.001 s and a standard deviation of 0.06 s for P waves and a mean of 0.005 s and a standard deviation of 0.25 s for S waves. For comparison, the absolute arrival-time errors of the pre-trained PhaseNet model compared with manual picks have a mean of 0.002 s and a standard deviation of 0.05 s for P waves and a mean of 0.003 s and a standard deviation of 0.08 s for S waves[19]. Although the differential time errors and absolute arrival-time errors can not be directly compared, the similar scales of these errors demonstrate that we can effectively transfer the high picking accuracy of the pre-trained PhaseNet model to DAS data.

## Applications to earthquake monitoring

The experiments above demonstrate that PhaseNet-DAS can effectively detect and pick P- and S-phase arrivals with few false positives, high sensitivity, and precise temporal accuracy. These automatic phase arrival-time measurements can be applied to many seismic studies such as earthquake monitoring and seismic tomography. Here, we further applied PhaseNet-DAS to earthquake monitoring. Following a similar workflow of earthquake detection using seismic networks[66], we applied PhaseNet-DAS to DAS data of 11,241 earthquakes in the earthquake catalogs of Northern California Seismic Network, Southern California Seismic Network, and Nevada Seismic Network within 5 degrees from two Long Valley DAS arrays (Fig. 5). These events were filtered based on an approximate scaling relation determined by Yin et al.[67]. Because of different sensor coverages between seismic networks and DAS cables, seismic signals from distant but small magnitude events are expected to be too small to be detected by DAS, the absolute number of earthquakes in the standard catalogs and those detected by DAS can not be directly compared. To evaluate the improvements from semi-supervised learning, we compared the magnitude and distance distributions of earthquakes detected by three models, PhaseNet, PhaseNet-DAS v1, and PhaseNet v2, in Fig. 4 and Fig. S3. PhaseNet-DAS significantly improves detection of both small magnitude events near the DAS array and large magnitude events at greater distances. We also plotted the approximate locations of these detected earthquakes determined by phase association (Fig. 5 and Fig. S4). The locations of events within the Long Valley caldera,

which are close to the DAS array, can be well-constrained using these automatic arrival-time measurements, while the earthquake locations become less constrained with increasing epicentral distances due to the limited azimuthal coverage of a single DAS array (Fig. S5). The physical limitation in azimuth and distance coverage could be addressed by combining seismic networks, deploying additional DAS arrays, or designing specific fiber geometries in future research.

Lastly, we evaluated PhaseNet-DAS on continuous data to demonstrate its potential applications in large-scale data mining and real-time earthquake monitoring. We applied PhaseNet-DAS to 180 h of continuous data from 2020/11/17 to 2020/11/25 using a 5000-channel × 200-s window sampled at 100 Hz without overlap. As PhaseNet-DAS is a fully convolutional network (Fig. 7) and the convolution operator is independent of input data size, it can be directly applied to various time lengths and channel numbers subject to the memory limitations of computational servers. The picked phase arrivals were associated using GaMMA in the same manner as above. Fig. S6 shows the detected and associated picks from three models: PhaseNet, PhaseNet-DAS v1, and PhaseNet-DAS v2. The results from these models show a good consistency, while PhaseNet-DAS proves more effective in detecting several times more picks. To assess the potential for false positive events, we compared the associated earthquakes with events in standard earthquake catalogs. The histograms of temporal earthquake frequency in Fig. S7 reveal a good correlation between events detected by the DAS array and seismic networks. In particular, for events within -0.5 degree of the DAS cable (Fig. S7c), we can observe that earthquake frequencies vary from over 80 events to no events per 6-hour period. Given the background noise generally does not change dramatically from day to day, this indicates that these detections are less likely to be false detections from noise sources such as traffic. In addition to the high correlation with the standard catalog, PhaseNet-DAS can detect 2–3 times more events using DAS alone, demonstrating the potential of combining fiber-optic networks to enhance the earthquake monitoring capability of conventional seismic networks. The entire processing time of the continuous DAS data (180 hours and 10,000 channels, 1.8 million channel-hours) was ~3.5 h using 8 GPUs (NVIDIA Tesla V100). The model prediction of PhaseNet-DAS is fast considering the substantial size of DAS data. Since the phase-picking task can be embarrassingly parallelized by segmenting DAS data into windows, the model prediction can be

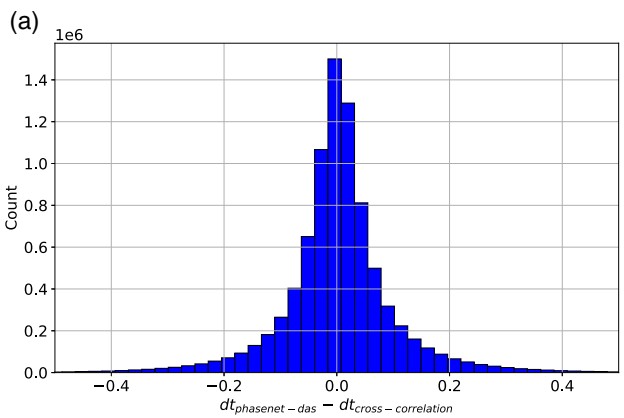

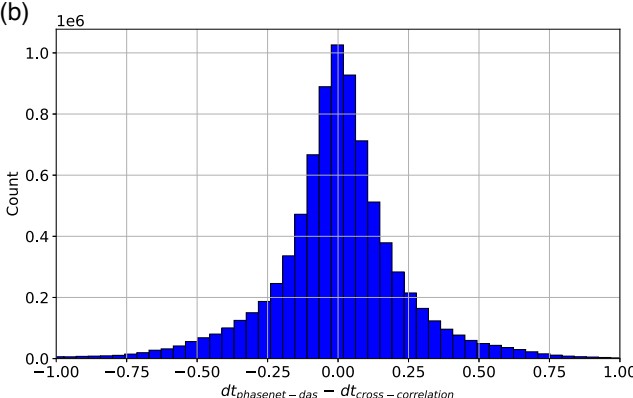

**Fig. 3 | Residuals of differential arrival-times picked by PhaseNet-DAS. a** P waves and **b** S waves. We first measure differential arrival-times of PhaseNet-DAS picks ($dt_\text{phasenet-das}$) and waveform cross-correlation ($dt_\text{cross-correlation}$) from selected event pairs. Then we calculate the residuals between these two differential arrival-times ($dt_\text{phasenet-das} - dt_\text{cross-correlation}$) to evaluate the accuracy of PhaseNet-DAS picks, assuming waveform cross-correlation measurement as the ground truth.

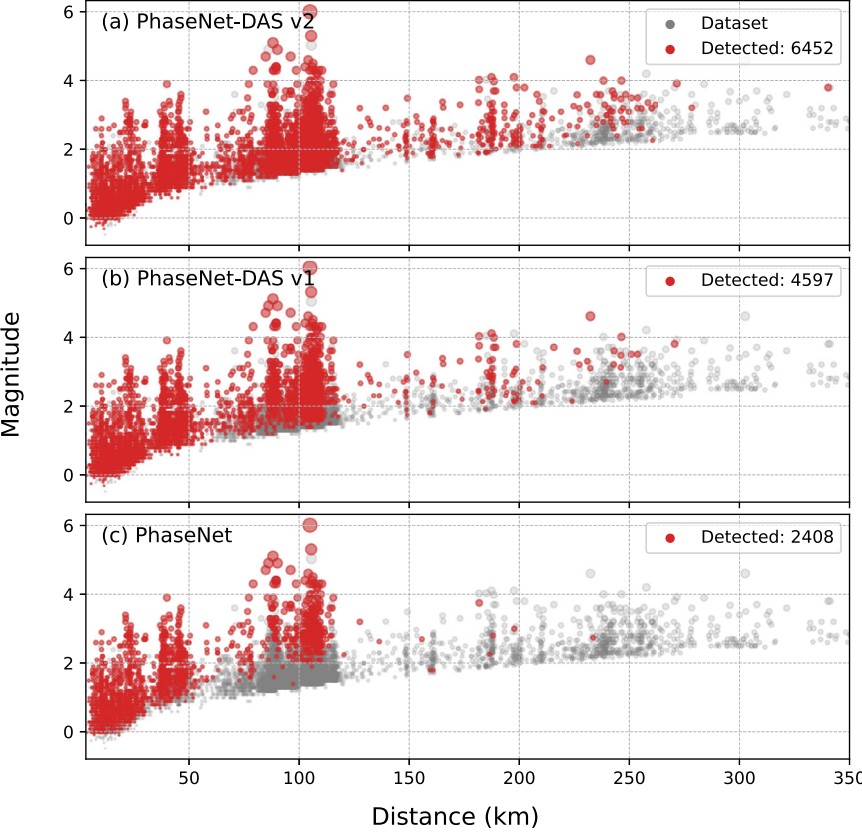

**Fig. 4 | Magnitude and distance distributions of earthquakes. a** PhaseNet-DAS v2, **b** PhaseNet-DAS v1, and **c** PhaseNet. The gray dots are earthquakes in standard earthquake catalogs within 3 degrees of the Long Valley DAS array. The red dots indicate the earthquakes that can be detected with more than 500 associated P and S picks. The PhaseNet-DAS v1 and v2 models are from the first and second iterations of semi-supervised learning (Fig. 6). The histogram of earthquake numbers is shown in Fig. S3.

further accelerated with additional GPUs for large-scale data mining tasks. The rapid prediction speed of PhaseNet-DAS also demonstrates its potential for real-time earthquake monitoring and earthquake early warning.

## Discussion

DAS enhances seismic observations by turning the existing fiber optic infrastructure into dense arrays of sensors, recording seismic waveforms with unprecedented spatial resolutions. Meanwhile, deep learning advances seismic data processing by transforming historical datasets into effective models for analyzing earthquake signals. PhaseNet-DAS attempts to combine these advantages to effectively detect and pick seismic phase arrivals in DAS data. The semi-supervised learning approach bridges the gap between two distinct data modalities of 1D conventional seismic waveforms and 2D DAS recordings. This approach addresses the challenge of lack of manual labels on DAS data, facilitating an efficient transfer of phase-picking capability from pre-trained deep learning models on 1D time series of

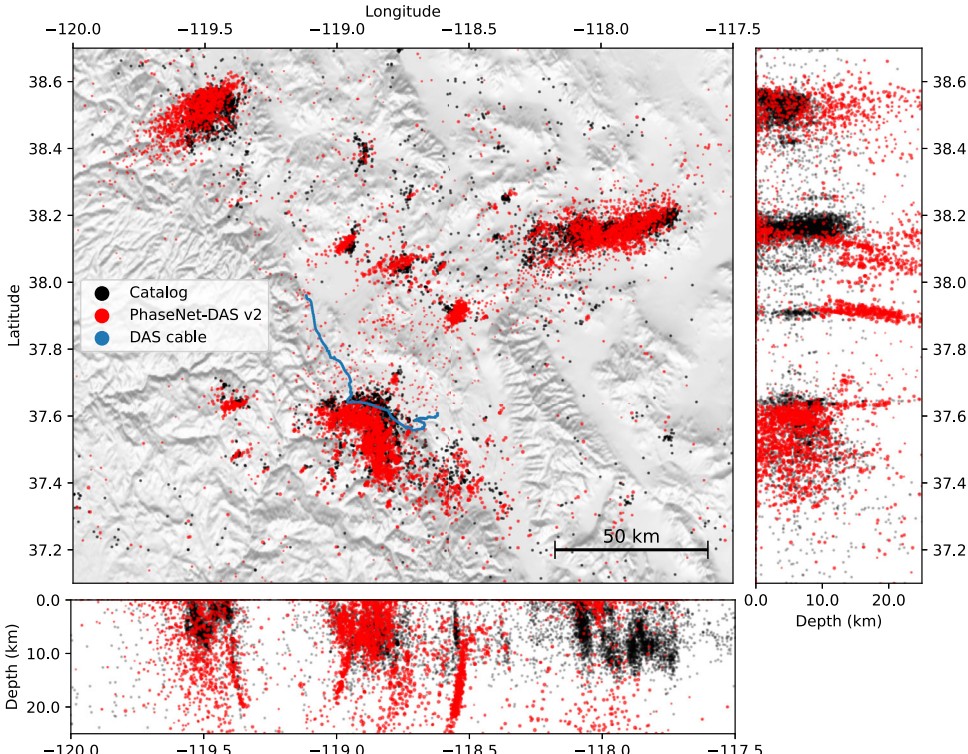

**Fig. 5 | Earthquake locations determined by phase arrival-times measured by PhaseNet-DAS.** The black dots are earthquakes in the standard earthquake catalogs. The red dots are earthquakes detected by the DAS arrays and the PhaseNet-DAS v2 model. Only the DAS events corresponding to a catalog event are shown. The results of PhaseNet and PhaseNet-DAS v1 models are shown in Fig. S4.

seismic data to new models designed for 2D spatio-temporal measurement of DAS data. In addition to earthquake monitoring, the PhaseNet-DAS model can be applied to other tasks such as seismic tomography and source characterization. It would be promising to explore whether the semi-supervised approach could also serve in developing deep learning models for other seismic signals in DAS data, such as detecting tremors[13,68] and picking first motion polarities[69] where large seismic archives are available.

Our experiments demonstrate the improvements from semi-supervised learning. PhaseNet-DAS, which is trained to pick phases across multiple channels of a DAS array, can effectively reduce false positive picks (Fig. 1 and Table S1), increase phase picks per event (Fig. S1), detect more low SNR events (Fig. 2, Fig. 4, and Fig. S3), and achieve a temporal precision similar to PhaseNet (Fig. 3). However, potential limitations of the current model should also be considered. While the semi-supervised learning approach addressed the challenge of the lack of manual labels for DAS data, the pseudo labels generated by the pre-trained PhaseNet model could potentially be subject to systematic bias, such as missing very weak first arrivals or confusing phase types using single-component data. In order to mitigate these biases, we adopted two approaches in this work. Firstly, we applied phase association to filter out inconsistent phase picks across channels. While the phase-picking step using PhaseNet only considers information from a single channel, the phase association step incorporates physical constraints across multiple channels, i.e., the phase type should be the same for nearby channels, and the phase arrival time should follow the time move-out determined by channel locations and wave velocities. Through phase association, we reduce the potential bias in pseudo labels of inaccurate phase time or incorrect phase types. Secondly, we added strong data augmentation to the training dataset to increase its size and diversity. For example, we superpose various real noises on the training dataset in order to make the model more sensitive to weak phase arrivals. Because the pseudo

labels are generated using data from high SNR events, sharp and clear first arrivals are less likely to be missed by PhaseNet. By superposing strong noise, we can make these arrivals similar to the cases of low SNR data from either small magnitude earthquakes or strong background noise, such as during traffic hours. Through such data augmentation, we can reduce the potential bias in pseudo labels of missing weak arrivals for low SNR events. Other approaches, such as employing waveform similarity, could also be considered to further reduce bias in pseudo labels. Incorporating regularization techniques, such as adding Laplacian smoothing between nearby channels to the training loss, could be another direction to reduce the effect of inconsistent labels and improve model performance in future research.

Another common challenge for deep learning is model generalization to new datasets, as the performance of deep neural networks is closely tied to the training datasets. The current PhaseNet-DAS model was trained and tested using only four DAS arrays in Long Valley and Ridgecrest, CA. The datasets are also formatted using a same temporal sampling of 100 Hz and a similar spatial sampling of ~10 m. These factors may limit the model's generalization to DAS arrays at different locations and/or with varying spatial and temporal sampling rates. However, because manual labels of historical seismic data are readily available at many locations, we can also apply the semi-supervised learning approach to train deep learning models for other DAS arrays or fine-tune the pre-trained PhaseNet-DAS models if limited DAS data is available.

In conclusion, with the deployment of more DAS instruments and the collection of massive DAS datasets, developing novel data processing techniques becomes a key direction in discovering signals and gaining insights from massive DAS data. Deep learning is widely applied in seismic data processing but has limited applications to DAS data due to the lack of manual labels for training deep neural networks. We explored a semi-supervised learning approach to pick P- and S-phase arrivals in DAS data without manual labels. We applied the pre-

trained PhaseNet model to generate noisy phase picks, used the GaMMA model to associate consistent picks as pseudo labels, and trained a new deep neural network model, PhaseNet-DAS, which is designed to utilize both temporal and spatial information of DAS data. The experiments demonstrate that PhaseNet-DAS can effectively detect P and S arrivals with fewer false picks, higher sensitivity to weak signals, and similar temporal precision compared to the pre-trained PhaseNet model. PhaseNet-DAS could be applied to earthquake monitoring, early warning, seismic tomography, and other seismic data analysis using DAS. The semi-supervised learning approach bridges the gap between limited DAS training labels and abundant historical seismic manual labels, facilitating future developments of deep learning models for DAS data.

## Methods

In this section, we discuss three components of applying deep learning to accurately pick phase arrival times in DAS data: the semi-supervised learning approach, the PhaseNet-DAS model, and the training dataset.

### Semi-supervised learning

We explore a semi-supervised learning approach to train a deep-learning-based phase picker using unlabeled DAS data. The procedure of the semi-supervised learning approach is summarized in Fig. 6.

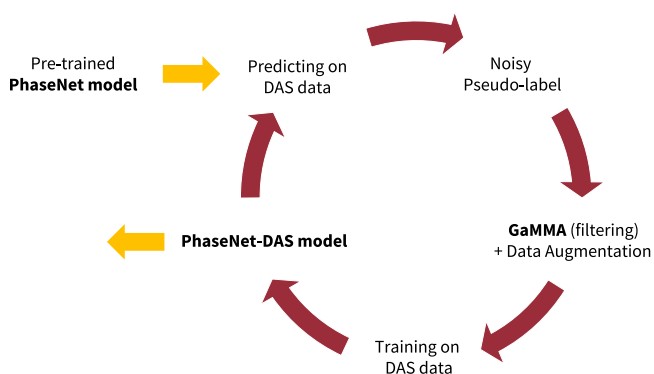

**Fig. 6 | The procedure of semi-supervised learning for training the PhaseNet-DAS model.** The PhaseNet model[19], which is pre-trained using a large dataset of seismic waveforms, is used to generate pseudo-labels on DAS data. This semi-supervised approach transfers the phase picking capability from PhaseNet to the new PhaseNet-DAS model designed for DAS recordings.

First, we train a deep-learning-based phase picker on three-component seismic waveforms using many analyst-labeled manual picks. Given the existence of several widely used deep-learning-based phase pickers[18–20], we directly reuse the pre-trained PhaseNet[19] model to omit retraining a deep-learning phase picker for conventional seismic data, which is not the focus of this work. Despite PhaseNet being trained on three-component seismic waveforms, it can also be applied to single-component waveforms because channel dropout (i.e., randomly zero-out one or two channels) is added as data augmentation[70].

Second, we apply the pre-trained PhaseNet model to pick P and S arrivals on each channel of a DAS array independently to generate noisy pseudo labels of P and S picks. While PhaseNet works well on channels with high signal-to-noise (SNR) ratios in DAS data, its accuracy is limited compared to that in seismic data (Fig. 1). For example, the model could detect many false picks due to strong anthropogenic noise in DAS data. The picked phase arrival times also have large variations between nearby channels, since each channel is processed individually.

Third, we apply the phase association method, Gaussian Mixture Model Associator (GaMMA)[57] to filter out false picks and build a DAS training dataset with pseudo labels. GaMMA selects only picks that fall within a narrow window of the theoretical arrival times corresponding to the associated earthquake locations. We set the time window size to 1 s in this study (Table S2). This hyperparameter can be adjusted to balance the trade-off between the quantity and quality of pseudo labels. A small window size results in a small training dataset with high-quality pseudo labels. Conversely, a large window size creates a large training dataset with potentially less accurate arrival times.

Last, we train a new deep-learning-based phase picker designed for DAS data. The model architecture is explained in the following section (Fig. 7). The training labels utilize the same Gaussian-shaped target function as proposed by Zhu and Beroza[19]:

$$P_P = e^{-\frac{(t-t_P)^2}{2\sigma^2}} \tag{1}$$

$$P_S = e^{-\frac{(t-t_S)^2}{2\sigma^2}} \tag{2}$$

$$P_N = \max(0, 1 - P_P - P_S) \tag{3}$$

where $t_P$ and $t_S$ are the arrival-times of P and S phase; $P_P$, $P_S$, and $P_N$ are the target functions for P-phase, S-phase, and Noise; $\sigma$ is the width of the Gaussian-shaped target function, which is used to account for

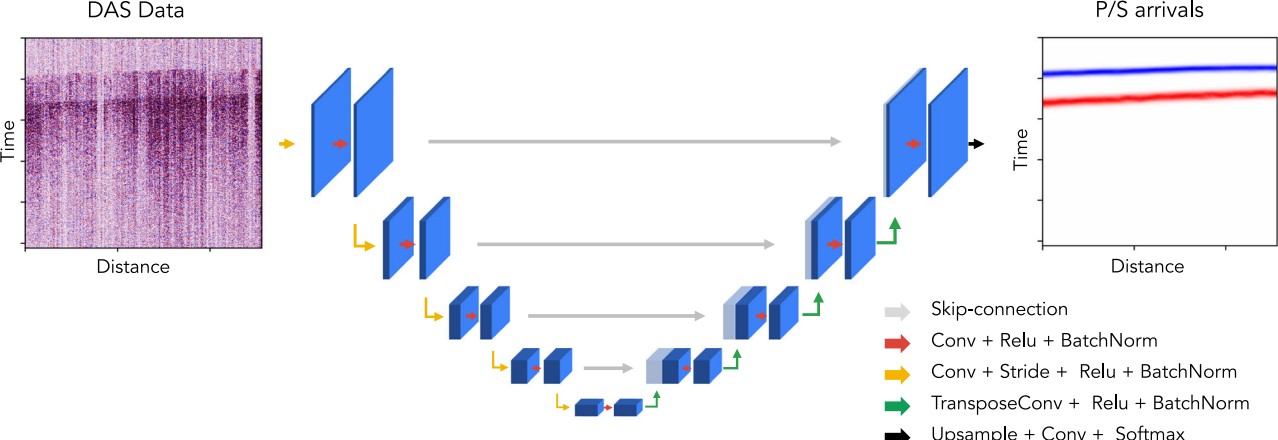

**Fig. 7 | The neural network architecture of PhaseNet-DAS.** We use the U-Net[73] architecture to consider spatial and temporal information of 2D DAS recordings. PhaseNet-DAS processes raw DAS data through four stages of downsampling and upsampling and a sequence of 2D convolutional layers and relu activation functions and predicts P and S phase arrivals in each channel of the DAS array, represented by blue and red lines respectively.

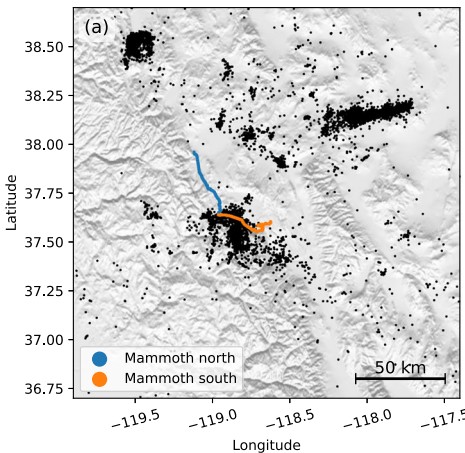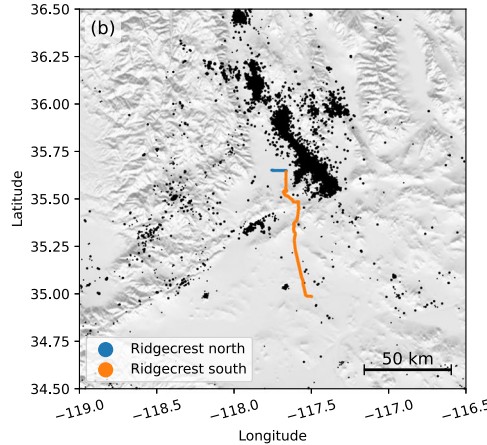

**Fig. 8 | Four DAS cables used in the training dataset. a** Long Valley and **b** Ridgecrest, CA. The blue and orange lines are the locations of the fiber-optic cables. The black dots are earthquakes in the standard earthquake catalogs used to build the training dataset.

uncertainties in phase arrival times similar to label smoothing commonly used in computer vision[71]. We set $\sigma$ to 1.5 s in this work. Because the pseudo labels are mostly picked on high SNR channels, a deep learning picker trained only on high SNR waveforms could generalize poorly to noisy waveforms. Data augmentation, such as superposing noise onto seismic events, can synthesize new training samples with noisy waveforms, significantly expand the training dataset, and improve model generalization on noisy DAS data and weak earthquake signals[45,72]. In addition to superposing noise, we add augmentations of randomly flipping data along the spatial axis, masking part of data, superimposing double events, and stretching (resampling) along the temporal and spatial axes.

By following these steps, we can automatically generate a large dataset of high-quality pseudo labels and train a deep neural network model on DAS data. We can further use this newly trained model to generate pseudo labels and train an improved model. This procedure can be repeated several times to enhance performance. In this study, we conducted two iterations using pseudo labels generated by PhaseNet and PhaseNet-DAS. We named the two resulting models as PhaseNet-DAS v1 and v2 for clarity. Future work could further optimize the number of iterations to enhance performance.

### Neural network model

The pre-trained PhaseNet model is a modified U-Net architecture[73] with 1D convolutional layers for processing 1D time series of seismic waveforms. DAS data, on the other hand, are 2D recordings of seismic wavefields with both spatial and temporal information. So the pre-trained PhaseNet model cannot utilize the spatial information from DAS's ultra-dense channels. In order to exploit both spatial and temporal information of 2D DAS data, we extend the PhaseNet model using 2D convolutional layers. The architecture of the PhaseNet-DAS model is shown in Fig. 7, which is similar to the original U-Net architecture[73]. In order to consider the high spatial and temporal resolution of DAS data, we use a larger convolutional kernel size (7 × 7) and a stride step (4 × 4) to increase the receptive field of PhaseNet-DAS[74]. We add the transposed convolutional layers for up-sampling[75], batch normalization layers[76], relu activation functions[77], and skip connections to the model. The semi-supervised approach does not require using the same neural network architecture as the pre-trained model, so that we can also use other advanced architectures designed for the semantic segmentation task, such as DeepLab[78], deformable ConvNets[79], and Swin Transformer[80]. In this work, we focus on exploring whether we can transfer the knowledge of seismic phase picking from seismic data to DAS data, so we keep a simple U-Net

architecture as PhaseNet. The exploration of optimal neural network architectures, e.g., transformer[22,80,81], for DAS data could be done in future research.

### Training data

We collected a large training dataset using four DAS cables in Long Valley and Ridgecrest, CA (Fig. 8). The Long Valley DAS array consists of two cables, each with a length of 50 km, 5000 channels, and a channel spacing of ~10 m[8,82–85], referred to as the Mammoth north and Mammoth south cables for clarity (Fig. 8a). The Ridgecrest DAS array consists of one short cable (10 km and 1250 channels) and one long cable (100 km and 10,000 channels), referred to as the Ridgecrest north and Ridgecrest south cables respectively (Fig. 8b). The cable locations are determined using a GPS-tracked moving vehicle developed by Biondi et al.[85]. We extracted event-based DAS records based on the standard catalogs of the Northern California Seismic Network, Southern California Seismic Network, and Nevada Seismic Network. Following the semi-supervised learning approach outlined above, we applied the pre-trained PhaseNet model to pick P and S arrivals in these extracted event data, applied the GaMMA model to associate picks, and kept the events with at least 500 P and S picks as the training datasets. In the first iteration using PhaseNet as the pre-trained model, we obtained a dataset of 1056 events and 1116 events from the Mammoth north and Mammoth south cables, and 597 events and 1430 events from the Ridgecrest north and Ridgecrest south cables respectively. In the second iteration using PhaseNet-DAS v1 as the pre-trained model, we obtained a dataset of 3405 events and 3437 events from the Mammoth north and Mammoth south cables, and 3590 events and 3311 events from the Ridgecrest north and Ridgecrest south cables respectively. Because we do not have manual labels as ground truth to evaluate the model performance, we only split each dataset into 90% training and 10% validation sets. We randomly selected training samples of 3072 × 5120 (temporal samples × spatial channels) and applied a moving window normalization to each channel. The moving window normalization, implemented using a convolutional operation with a window size of 1024 and a stride step of 256, removes the mean and divides by the standard deviation within a fixed window size, making it independent of input data length. Coupled with the fully convolutional network architecture of PhaseNet-DAS, the model can be applied to flexible length of continuous data. We trained PhaseNet-DAS using the AdamW optimizer and a weight decay of 0.1[86,87], an initial learning rate of 0.01, a cosine decay learning rate with linear warm-up[88], a batch size of 8, and 10 training epochs. Future work could further explore optimal hyperparameters to enhance performance.

## Data availability

The example dataset of the Ridgecrest north cable is available at: https://doi.org/10.57967/hf/0962. These examples are extracted from the public Ridgecrest DAS dataset hosted under the SCEDC Earthquake Data AWS Public Dataset (https://scedc.caltech.edu/data/getstarted-pds.html). The other DAS datasets used for training and testing are available upon request from Zhongwen Zhan (zwzhan@caltech.edu).

## Code availability

The pre-trained model of PhaseNet is available at https://ai4eps.github.io/PhaseNet/. The model of GaMMA is available at https://ai4eps.github.io/GaMMA/. The source code and pre-trained model of PhaseNet-DAS is available at https://ai4eps.github.io/EQNet/[89].

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

## Acknowledgements
We would like to thank the reviewers and editors for their insightful comments and constructive suggestions. We would like to thank James Atterholt for his assistance in building the training dataset. We would like to thank James Atterholt, Qiushi Zhai, Yan Yang, and Jiaqi Fang for their constructive discussions. We would also like to thank the California Broadband Cooperative for fiber access for the Distributed Acoustic Sensing array used in this experiment. We would like to thank OptaSense for the support provided for this calibration experiment. In particular, we thank Martin Karrenbach, Victor Yartsev, and Vlad Bogdanov. This study is funded by the Gordon Moore Foundation (Z.Z.), the National Science Foundation (NSF) through the Faculty Early Career Development (CAREER) award number 1848166 (Z.Z.), and the United States Geological Survey Earthquake Hazards Program award number G22AP00067 (Z.Z.).

## Author contributions
W.Z. developed and implemented the algorithm, conducted the experiments and analysis. E.B., Z.R., and Z.Z. co-designed the study. J.L. conducted the picking time error analysis. J.L. and J.Y. built the DAS dataset and tested the model. Z.R. and Z.Z. advised the project. All authors contributed to writing and editing the manuscript.

## Competing interests
The authors declare no competing interests.
