## [Peer Review File · Nature Communications]

Seismic Arrival-time Picking on Distributed Acoustic Sensing Data using Semi-supervised LearningREVIEWER COMMENTS

Reviewer #1 (Remarks to the Author):

The paper “Seismic Arrival-time Picking on Distributed Acoustic Sensing Data using Semi-supervised Learning” by Zhu et al describes a deep learning method for seismic phase arrival time picking and event detection on DAS data. To this end, the authors generalize the PhaseNet approach to DAS data. The key contribution of the manuscript is a novel, semi-supervised training strategy for the model, combining noisy labels with a phase association step for external validation. This strategy alleviates the problem of missing manually annotated phase arrival labels for DAS.

With the quickly growing adoption of DAS measurements, providing a sensitive and high-performing method for detecting earthquakes in DAS data is a timely and important topic. The provided method is sound and well suited for the task and, if published with a sufficiently user-friendly implementation, will be of great interest to practitioners. Nonetheless, I have two larger points of concern about the current manuscript.

First, the metrics for event detection seem rather poor to me. At local to regional distances, already an offset of 3 s in origin time is considerable. At 15 s, it is questionable if this is still the same event. In addition, there are 25 % of events not associated with any cataloged event. Are these additional detections or actually false positives? On the other side, this also implies that there are a substantial number of events that have not been detected with the DAS. Looking at Figure 4, a lot of these seem to be around the detection limit, but there are also several M3.5+ events at distances of around 100 km that were missed and should definitely be visible on the DAS. To understand the usefulness of the model, I think it is essential to understand these missed/falsed detections. This is critical for users aiming to build catalogs with the method, as it might distort the catalogs and subsequent analysis on the catalogs. Related to this point, I think to really prove the necessity of the method for catalog generation, it should be compared to a regular PhaseNet. Looking at the examples in Figure 1, PhaseNet picks are pretty noisy, but the events are still clearly identifiable. In particular, selecting good labels through association shows that GaMMA can actually identify the events from these picks. Therefore, I suggest to also compare the catalog obtained from directly applying GaMMA to the PhaseNet picks with the GaMMA + PhaseNet-DAS catalog.

Second, as indicated in Figure 5, a key advantage of the method is that it can be iterated. However, according to line 280, this is actually not done. Looking at the PhaseNet predictions in panel 1, this suggests that the model is trained on very noisy labels and that retraining would considerably improve the model. Therefore I think at least one iteration of this association loop starting with a PhaseNet-DAS model should be conducted.

As a last point, I strongly encourage the authors to deposit their data in a data repository with a DOI. Making data available from the authors upon request increases the risk of data becoming inaccessible in the future and is not in line with open science practices.

In conclusion, I think this manuscript tackles an important and timely topic and will be a valuable contribution to the community. However, there are major points that from my perspective need to be addressed before publication. In addition, I list a collection of minor points below. I encourage the authors to address these points and think this work will be very useful in future studies of DAS data.

Minor:

- The references are numbered in-text but sorted alphabetically (and without numbers) in the bibliography. This makes cross-checking of the references for the reviewers almost impossible.
- Introduction: I think it should be mentioned that another advantage of DAS is that it can be deployed offshore pretty easily.
- Line 40ff: There should be a note that the high data volume is also a key limiting factor for DAS processing.
- Lines 48ff: The reference to template matching seems a bit unrelated to me at this point. While the application of STA/LTA to DAS is pretty straightforward and this method has been used, I'm not aware that the same has been done with template matching.
- Line 79f: Can you provide references that these methods are not effective for DAS data? I admit, this is likely difficult as null results are rarely published but, if possible, would be a helpful resource for readers.
- I think the proposed model does not really fit the noisy student method framework. The key part of the training strategy is using the association step as an outside validation signal. Such an outside validation is not present in the noisy student method. Personally, I'd call this distantly-supervised learning but machine learning terminology is far from unique here.
- Line 151f: Please provide the number of channels to put the numbers of picks in context.
- From personal experience, GaMMA runtimes grow strongly with the number of picks in each cluster (of the initial DBscan step). With the enormous number of picks in this dataset, how does GaMMA perform in terms of runtime?
- For the generated catalog, was PhaseNet-DAS trained on this sequence or is this an independent training set.
- Line 169f: Please provide the employed overlap between windows. As pickers are unfortunately not invariant to shifting, I think a certain overlap between windows is necessary and this value can considerably impact runtime.
- Lines 169 and 213: There are two different sampling rates (100 Hz and 200 Hz) provided. Does this mean that different sampling rates have been used in training and application?
- Figure 4: As far as I understood from the text, only the DAS events corresponding to a catalog event are shown. Could you please specify this in the caption. In addition, it could maybe be indicated visually. At first glance, it seemed like the DAS had detected substantially more events, including large ones.
- Please describe if and how hyperparameter tuning has been performed.
- How are the labels for PhaseNet-DAS encoded? I'd assume the encoding are probability curves similar to PhaseNet but this should be explicitly specified.
- For applying GaMMA, the location of each channel of the DAS is required. However, DAS channel locations are generally not perfectly known as the cables might not be absolutely straight. Please provide a discussion on how the channel locations were determined and if the uncertainty has an effect on the model.

Reviewer #2 (Remarks to the Author):

This manuscript describes the development of a seismic phase picker based on deep-learning approaches that can be applied to Distributed Acoustic Sensing (DAS) data. There is little doubt that such a tool is useful, and it seems indeed that the authors produced a working solution.

The manuscript is well written but in a somewhat exuberant language that gives, from the beginning, the impression that the authors may be trying to sell something. The organisation of the manuscript, with excessive focus on results but rather sparse information on the actual developments, further enhances this impression.

I read this manuscript several times, with a few days between, and now see three major classes of concerns:

INNOVATION:

I do understand that deep learning is currently a big hype and that it is tempting, especially for young researchers, to become part of this. However, the days where the mere application of some deep-learning model was science, are over. With numerous easily usable programming tools, the training of a neural network (deep or not; this is just somewhat hollow semantics) has become an easy task for undergraduate students. After all, it is what it is: fitting the coefficients of some functional form to a collection of data.

This said, it is not clear how exactly the authors go significantly beyond just training yet another deep neural network? What is the innovation that goes beyond the obvious?

Admittedly, I do not find these questions easy to answer. The only problem that the authors seem to address is the absence of labelled training datasets for DAS data. For this, they exploit existing single-trace phase pickers, the results of which are filtered by a phase association algorithm. Essentially, this ensures that signals without spatial coherence do not end up being used as pseudo-labels for the new network training.

In my opinion, the whole innovation in this work is precisely this filtering step. Nothing more. This is not to say that there is no innovation! However, its magnitude does not match the grandiose tone of the manuscript, and I also do not think this is sufficient for a high-profile publication.

METHOD:

Looking at this filtering step from some distance, I think it becomes clear that the rest of the proposed method, i.e., the training of a new network, has actually become redundant! After the filtering, i.e., the coherence-based cleaning of the single-trace picks, the problem of extracting seismic phases from the DAS data is actually already solved! The output of the filter is already what we need, and there is no

obvious need to then build a new interpolation function(neural network) on top of that.

Hence, the whole purpose of half the method is actually not clear.

I must admit that I may not have understood the method 100 %. However, in this case, I strongly suspect that a majority of the readers would not understand it either. Again, this goes hand in hand with the peculiar presentation style.

UNCERTAINTIES:

The authors' method also seems to produce earthquake locations. However, it seems like they are merely dots on a map. For many decades, earthquake locations have been reported with uncertainties, for obvious reasons.

Here, it is not clear to me where these uncertainties are and how the method could actually produce them. In case it can't, the method basically has no value for quantitative science. (It still makes impressive pictures.) Hence, this part needs some serious work.

In summary, I do not think that there is sufficient innovation and creativity for a high-profile publication, and not enough technical information and detail for a more method-oriented journal.

Reviewer #3 (Remarks to the Author):

The paper presents a semi-supervised deep learning approach (PhaseNet-DAS) to pick seismic phase arrival times in distributed acoustic sensing (DAS) data. The study proposes the use of pre-trained PhaseNet to generate noisy pseudo-labels of P and S picks from individual DAS channels. Next, the study uses a phase association method (Gaussian Mixture Model Associator (GaMMA)) to distinguish noise and seismic arrivals from the pseudo-labels, essentially clustering respective phases into groups. Data augmentation (random flipping) is also added to increase the diversity of the dataset. By following this methodology, a large number of training datasets can be generated, and the quality of the dataset can be refined by repeating the procedure several times if desired. Consequently, the study proposes PhaseNet-DAS, which takes in two-dimensional (2D) DAS data, and predicts the relevant 2D P and S arrivals.

This study presents a major contribution to the DAS geophysics community. DAS is often clouded with inherent noise due to its highly sensitive and dense sensors. Existing machine learning (ML) pickers only work best on individual traces, which is likely to predict inconsistent phase picks when applied on 2D DAS data. As evident in Figure 1, PhaseNet-DAS seems to be able to predict the P and S picks with

minimal noise. Furthermore, the semi-supervision nature of this study means that less manual work is needed to label 2D DAS signals. This contribution is significant because manually labeling 2D DAS signals can be highly laborious at times.

Comments/questions:

As DAS is not only used to monitor earthquakes, is PhaseNet-DAS limited to only detecting earthquakes? Can PhaseNet-DAS predict non-earthquake signals captured by DAS? Some examples of other seismic signals include vehicular traffic, mining blasts, thunderstorms, and microseismicity. I suggest discussing the limitations and/or feasibility of PhaseNet-DAS on detecting non-earthquake signals.

For the deep learning training, how are the input 2D datasets (1024x1024) cut? What about cases when a cut dataset contains only S arrivals? Would this affect the training and prediction?

We thank the editor and reviewers for your valuable feedback and constructive suggestions. We performed additional experiments and made revisions to our manuscript, including three major changes: First, we implemented a second iteration of training using the semi-supervised learning approach. For clarity, we named the first iteration as PhaseNet-DAS v1, which is trained using noisy labels generated by PhaseNet, and named the second iteration as PhaseNet-DAS v2, which is trained using noisy labels generated by PhaseNet-DAS v1. Second, we compared the performance of PhaseNet, PhaseNet-DAS v1, and PhaseNet-DAS v2. We associated the picks from each model using the same phase associator GaMMA, and compared the association rate and detected earthquake numbers. Last, we added more metrics to show the improvements after semi-supervised learning, such as the number of picks per event, time residuals of picks, and number of events in relation to SNR and distance. Below we describe the changes that were made in response to reviewers' comments.

REVIEWER COMMENTS

Reviewer #1 (Remarks to the Author):

The paper “Seismic Arrival-time Picking on Distributed Acoustic Sensing Data using Semi-supervised Learning” by Zhu et al describes a deep learning method for seismic phase arrival time picking and event detection on DAS data. To this end, the authors generalize the PhaseNet approach to DAS data. The key contribution of the manuscript is a novel, semi-supervised training strategy for the model, combining noisy labels with a phase association step for external validation. This strategy alleviates the problem of missing manually annotated phase arrival labels for DAS.

A: Thank you for recognizing the contributions of our work.

With the quickly growing adoption of DAS measurements, providing a sensitive and high-performing method for detecting earthquakes in DAS data is a timely and important topic. The provided method is sound and well suited for the task and, if published with a sufficiently user-friendly implementation, will be of great interest to practitioners. Nonetheless, I have two larger points of concern about the current manuscript.

A: We will release the source code and the pre-trained model with the paper to make it easy for applying our model and re-training models for new DAS datasets. We also clarified the implementation details mentioned in the following comments.

First, the metrics for event detection seem rather poor to me. At local to regional distances, already an offset of 3 s in origin time is considerable. At 15 s, it is questionable if this is still the same event. In addition, there are 25 % of events not associated with any cataloged event. Are these additional detections or actually false positives? On the other side, this also implies that there are a substantial number of events that have not been detected with the DAS. Looking at

Figure 4, a lot of these seem to be around the detection limit, but there are also several M3.5+ events at distances of around 100 km that were missed and should definitely be visible on the DAS. To understand the usefulness of the model, I think it is essential to understand these missed/falsed detections. This is critical for users aiming to build catalogs with the method, as it might distort the catalogs and subsequent analysis on the catalogs.

Related to this point, I think to really prove the necessity of the method for catalog generation, it should be compared to a regular PhaseNet. Looking at the examples in Figure 1, PhaseNet picks are pretty noisy, but the events are still clearly identifiable. In particular, selecting good labels through association shows that GaMMA can actually identify the events from these picks. Therefore, I suggest to also compare the catalog obtained from directly applying GaMMA to the PhaseNet picks with the GaMMA + PhaseNet-DAS catalog.

A: Thanks for this constructive suggestion. We have added additional results to address these comments.

1. We have included results of both PhaseNet and PhaseNet-DAS in our analysis. We updated Figure 4 to include catalogs generated using PhaseNet+GaMMA and PhaseNet-DAS+GaMMA. We added Figure S3 to show the change of detected earthquake number with distance. We also added Figure S4 to show earthquake locations using PhaseNet+GaMMA similar to Figure 5 using PhaseNet-DAS+GaMMA. The comparison helps to demonstrate the improved earthquake detection performance.

2. The large origin time error results from its trade-off with earthquake location and velocity errors. The limited azimuth coverage of DAS leads to poor constraints on earthquake location (Figure S5). If the earthquake locations are inaccurate, the origin time will also have a large error. As a result, it is not easy to directly compare earthquake catalogs generated by a seismic network and DAS. We added more explanation in the revision: L175 “Because of different sensor coverages between seismic networks and DAS cables, seismic signals from distant but small magnitude events are expected to be too small to detect, so the absolute number of earthquakes in the standard catalogs and detected by DAS can not be directly compared. To evaluate the improvements from semi-supervised learning, we compare the magnitude and distance distributions of earthquakes detected by three models, PhaseNet, PhaseNet-DAS v1, and PhaseNet v2 in Figure 4 and Figure S3. PhaseNet-DAS significantly improves the detection of both small magnitude events near the DAS array and large magnitude events at greater distances.”

We have also added an analysis of the time residuals of phase picks by comparing them with the theoretical phase arrival times based on the true earthquake origin times and locations (Figure S2). As suggested by the reviewer, most picks within ~100 km have small time residuals within a few seconds, while the time residuals become much larger as the epicenter distance increase due to the differences between the true 3D velocity model and the 1D velocity model we used.

3. Most additional detections come from multiple events in the same window, and some may inevitably be false positives. We select an event window of 120s to build our event dataset. For

earthquake swarms in Long Valley and aftershock sequences in Ridgecrest, there may include double or more events in one window. We can see several picks with large negative or positive time residuals at some distances in Figure S2.

The missed detections are mostly due to the low SNRs. We added Figure 2 to show how the detected earthquake number change with SNRs. Because the DAS recordings are not only related to distance, they are also affected by wave incident angle, coupling, local structures, and traffic noise level as the fiber-optic cables are mostly deployed along roads. We can see most of the missed events are around SNR = 0 dB in Figure 2. PhaseNet-DAS has a better performance than PhaseNet at low SNR. We have added more discussion about the detection performance: L129 “The PhaseNet model pre-trained on seismic data can detect high SNR events, but struggle with low SNR events on DAS (Figure 2). After re-training using semi-supervised learning on DAS data, the PhaseNet-DAS model significantly improves detections of low SNR events. PhaseNet-DAS v1 detects 2-5 times more events than PhaseNet across four DAS arrays, and PhaseNet-DAS v2 enhances detection sensitivity by an additional 25% - 50% compared to PhaseNet-DAS v1 (Figure 2).”

Second, as indicated in Figure 5, a key advantage of the method is that it can be iterated. However, according to line 280, this is actually not done. Looking at the PhaseNet predictions in panel 1, this suggests that the model is trained on very noisy labels and that retraining would considerably improve the model. Therefore I think at least one iteration of this association loop starting with a PhaseNet-DAS model should be conducted.

A: Thanks for the suggestion. We added another iteration and reported both the performance of PhaseNet-DAS v1 and v2.

As a last point, I strongly encourage the authors to deposit their data in a data repository with a DOI. Making data available from the authors upon request increases the risk of data becoming inaccessible in the future and is not in line with open science practices.

A: We deposit the extracted event data of the public Ridgecrest (north) DAS dataset under the DOI: <https://doi.org/10.57967/hf/0962>. The continuous DAS data can be accessed from the SCEDC Earthquake Data AWS Public Dataset (<https://scedc.caltech.edu/data/getstarted-pds.html>). Because other datasets are being used by undergoing projects in the group, we can not share them publicly at the current moment.

In conclusion, I think this manuscript tackles an important and timely topic and will be a valuable contribution to the community. However, there are major points that from my perspective need to be addressed before publication. In addition, I list a collection of minor points below. I encourage the authors to address these points and think this work will be very useful in future studies of DAS data.

A: Thank you for recognizing our contribution to the community and helping us improve our manuscript. We will open-source code and model to facilitate more applications of DAS to earthquake studies.

Minor:

- The references are numbered in-text but sorted alphabetically (and without numbers) in the bibliography. This makes cross-checking of the references for the reviewers almost impossible.

A: Corrected.

- Introduction: I think it should be mentioned that another advantage of DAS is that it can be deployed offshore pretty easily.

A: Good point. Added L37 “Furthermore, DAS is suitable for deployment and maintenance in challenging environments such as offshore locations, glaciers, and boreholes.”

- Line 40ff: There should be a note that the high data volume is also a key limiting factor for DAS processing.

A: Good point. Added L42 “... due to the lack of effective algorithms for detecting earthquakes and picking phase arrivals, coupled with the high data volume recorded by over thousands of channels.”

- Lines 48ff: The reference to template matching seems a bit unrelated to me at this point. While the application of STA/LTA to DAS is pretty straightforward and this method has been used, I’m not aware that the same has been done with template matching.

A: We have added two references of applications of template matching to DAS:

Li, Z., & Zhan, Z. (2018). Pushing the limit of earthquake detection with distributed acoustic sensing and template matching: A case study at the Brady geothermal field. Geophysical Journal International, 215(3), 1583-1593.

Li, Z., Shen, Z., Yang, Y., Williams, E., Wang, X., & Zhan, Z. (2021). Rapid response to the 2019 Ridgecrest earthquake with distributed acoustic sensing. AGU Advances, 2(2), e2021AV000395.

- Line 79f: Can you provide references that these methods are not effective for DAS data? I admit, this is likely difficult as null results are rarely published but, if possible, would be a helpful resource for readers.

A: Few studies have been conducted on phase picking on DAS data so far. We think the different data formats between seismic and DAS data is a challenge for transfer learning, while achieving high temporal accuracy is difficult for self-supervised learning. We think another potential solution for future research is to use self-supervised learning to train a large model first and fine-tune it on a small set of manually labeled dataset to achieve a good picking accuracy.

- I think the proposed model does not really fit the noisy student method framework. The key part of the training strategy is using the association step as an outside validation signal. Such an

outside validation is not present in the noisy student method. Personally, I'd call this distantly-supervised learning but machine learning terminology is far from unique here.

A: Thanks for this suggestion. We reference the Noisy Student method as an example of semi-supervised learning, since our training approach is similar to it. Our approach is also a type of semi-supervised learning using pseudo labels. In semi-supervised learning, key steps lie in filtering of pseudo labels, curation and augmentation of unlabeled dataset. For seismic data, phase association is a unique and effective way for filtering (injecting supervision information). For images, the Noisy Student method also applied multiple criteria for filtering: "We then perform data filtering and balancing on this corpus. ... We then select images that have confidence of the label higher than 0.3. For each class, we select at most 130K images that have the highest confidence. Finally, for classes that have less than 130K images, ..."

Xie, Q., Luong, M. T., Hovy, E., & Le, Q. V. (2020). Self-training with noisy student improves imagenet classification. In Proceedings of the IEEE/CVF conference on computer vision and pattern recognition

- Line 151f: Please provide the number of channels to put the numbers of picks in context.

A: We have added Figure S1 to show the number of picks per event. We can see the number of picks significantly increases for each detected event after re-training.

- From personal experience, GaMMA runtimes grow strongly with the number of picks in each cluster (of the initial DBscan step). With the enormous number of picks in this dataset, how does GaMMA perform in terms of runtime?

A: The implementation of DBSCAN in scikit-learn is pretty efficient. For associating 15,974 picks, GaMMA takes 9s on our server using a CPU of 64 cores.

- For the generated catalog, was PhaseNet-DAS trained on this sequence or is this an independent training set.

A: The catalog is generated on the same sequence. Our task is a bit different from supervised training, which has a clear test dataset for comparison. As shown in Table 1, the number of picks (pseudo labels) from PhaseNet is about 4.2M + 5.3M on the two DAS cable in Long Valley, while the model after training detects 35.9M + 40.2 picks. These new picks are outside of those used for training. But there could be a risk of overfitting on our DAS cables, since these new picks come from nearly but noisier channels. We will open source the model, so that people can test on other DAS cables. We also further expand this point in discussion: L242 "The current PhaseNet-DAS model was trained and tested using four DAS arrays in Long Valley and Ridgecrest, CA. The datasets are also formatted using a same time sampling of 100 Hz and a similar spatial sampling of ~10 m. These factors may limit the model's generalization to DAS arrays at different locations and/or with varying spatial and temporal sampling rates. "

- Line 169f: Please provide the employed overlap between windows. As pickers are unfortunately not invariant to shifting, I think a certain overlap between windows is necessary and this value can considerably impact runtime.

A: Because we are using a fully convolutional network architecture, we do not cut sliding-windows. L337 “We randomly selected training samples of 3072x5120 (temporal time x spatial size) and applied a moving window normalization to each channel. The moving window normalization, implemented using a convolutional operation with a window size of 1024 and stride step of 256, removes the mean and divides by the standard deviation within a fixed window size, making it independent of input data length. Coupled with the fully convolutional network architecture of PhaseNet-DAS, the model can be applied to flexible lengths of continuous data.”

• Lines 169 and 213: There are two different sampling rates (100 Hz and 200 Hz) provided. Does this mean that different sampling rates have been used in training and application?

A: The raw DAS is recorded at 200Hz, so we used the raw sampling rate in the speed test. The training and earthquake detection performance are both done at 100Hz

• Figure 4: As far as I understood from the text, only the DAS events corresponding to a catalog event are shown. Could you please specify this in the caption. In addition, it could maybe be indicated visually. At first glance, it seemed like the DAS had detected substantially more events, including large ones.

A: That is correct. We have updated the caption.

• Please describe if and how hyperparameter tuning has been performed.

A: We keep a setting similar to PhaseNet and did not apply any hyperparameter tuning. We have added this point in the revision:

L316 “In this work, we focus on exploring whether we can transfer the knowledge of seismic phase picking from seismic data to DAS data, so we keep a simple U-Net architecture as PhaseNet. The exploration of the best neural network architectures, e.g., transformer, for DAS data can be done in future research.”

L345 “Because the quality and size of the training datasets have a greater impact on our problem than hyperparameters, we do not tune these training hyperparameters.”

• How are the labels for PhaseNet-DAS encoded? I’d assume the encoding are probability curves similar to PhaseNet but this should be explicitly specified.

A: We have explicitly specified the training labels now: The training labels are the same Gaussian-shaped target function proposed by Zhu and Beroza (2019):

$$P_P = e^{-\frac{(t-t_P)^2}{2\sigma^2}}$$

$$P_S = e^{-\frac{(t-t_S)^2}{2\sigma^2}}$$

$$P_N = 1 - P_P - P_S$$

where t_P and t_S are the arrival-times of P and S phase, P_P , P_S , and P_N are the target functions for P-phase, S-phase, and Noise, and σ is the width of the Gaussian-shaped target function, which is used to account for uncertainty in phase arrival times similar to label smoothing used in computer vision.

- For applying GAMMA, the location of each channel of the DAS is required. However, DAS channel locations are generally not perfectly known as the cables might not be absolutely straight. Please provide a discussion on how the channel locations were determined and if the uncertainty has an effect on the model.

A: Biondi et al. (2023) developed a method to determine the DAS channel location pretty accurately using a GPS-Tracked Moving Vehicle. The channels locations of the four DAS arrays used in this study are determined using the proposed method. We have added this information.

In general, the channel locations are pretty accurate. The same DAS cable is also used for seismic tomography, currently under review. The earthquake location uncertainty is impacted by multiple factors is beyond the scope of this study. The limited azimuth coverage of a single DAS array is a major factor, as shown in Figure S5.

Biondi, E., Zhu, W., Li, J., Williams, E., & Zhan, Z. (2022, December). Fiber seismic tomography of the Long Valley Caldera. In AGU Fall Meeting Abstracts (Vol. 2022, pp. V33A-04).

Biondi, E., Wang, X., Williams, E. F., & Zhan, Z. (2023). Geolocalization of Large-Scale DAS Channels Using a GPS-Tracked Moving Vehicle. Seismological Society of America, 94(1), 318-330.

Reviewer #2 (Remarks to the Author):

This manuscript describes the development of a seismic phase picker based on deep-learning approaches that can be applied to Distributed Acoustic Sensing (DAS) data. There is little doubt that such a tool is useful, and it seems indeed that the authors produced a working solution.

The manuscript is well written but in a somewhat exuberant language that gives, from the beginning, the impression that the authors may be trying to sell something. The organisation of the manuscript, with excessive focus on results but rather sparse information on the actual developments, further enhances this impression.

I read this manuscript several times, with a few days between, and now see three major classes of concerns:

A: Thank you for taking the time to read our manuscript carefully. We have added more results based on your comments and revised the manuscript accordingly.

INNOVATION:

I do understand that deep learning is currently a big hype and that it is tempting, especially for young researchers, to become part of this. However, the days where the mere application of some deep-learning model was science, are over. With numerous easily usable programming tools, the training of a neural network (deep or not; this is just somewhat hollow semantics) has become an easy task for undergraduate students. After all, it is what it is: fitting the coefficients of some functional form to a collection of data.

This said, it is not clear how exactly the authors go significantly beyond just training yet another deep neural network? What is the innovation that goes beyond the obvious?

A: Thanks for these critiques. We think there are three major innovations of our work that goes beyond training yet another neural network. First, our work addresses the challenge of lack of manual labels for DAS data. We agree with the reviewers' comment, training a neural network is now an easy task even for undergraduate students. But there are no phase picking models trained for DAS data so far. The main challenge is not training neural networks but the lack of manual labels for training. The semi-supervised learning approach we proposed is an effective way to transfer the phase picking capability from seismic data to the new DAS data. Second, as the other two reviewers' have commented, DAS is a rapidly developing technology for earthquake monitoring, but there are no reliable earthquake detection and phase picking algorithm for DAS data. To our knowledge, this is the first phase detecting/picking model developed for DAS data. The picker already been used in several related studies of tomography, focal mechanism, fault zone imaging, earthquake magnitude estimation in our group. We will also open source our model to be used by the community, as we did in our previous works. Last, PhaseNet-DAS works significantly better than PhaseNet. Because PhaseNet is not designed for DAS and can not use spatial information between channels, so we did not include the comparison in our initial draft. We expanded the revision with the comparison with PhaseNet. Deep learning models such as PhaseNet only works on high SNR channels, which is also the basis of the possibility of semi-supervised learning from seismic data to DAS data. After training, PhaseNet-DAS has higher association rate (much fewer false picks) (Table 1), more picks per events (picking phase on noisy channels) (Figure S1), more detected events (sensitive to low SNR events) (Figure 2, Figure S3 and S4).

Admittedly, I do not find these questions easy to answer. The only problem that the authors seem to address is the absence of labelled training datasets for DAS data. For this, they exploit existing single-trace phase pickers, the results of which are filtered by a phase association algorithm. Essentially, this ensures that signals without spatial coherence do not end up being used as pseudo-labels for the new network training.

In my opinion, the whole innovation in this work is precisely this filtering step. Nothing more. This is not to say that there is no innovation! However, its magnitude does not match the

grandiose tone of the manuscript, and I also do not think this is sufficient for a high-profile publication.

A: Thanks for recognizing our innovation in addressing the absence of labeled training datasets for DAS data using semi-supervised learning. Lack of training labels for DAS is a major block for applying machine learning to DAS data. Although many papers on applications of DAS to earthquake studies are published, the applications of machine learning to DAS are limited. The semi-supervised learning approach we develop makes it possible to re-use many manual labels on seismic data and train machine learning models for DAS data. This approach can also be applied to other seismic tasks on DAS data such as picking first-motion polarity. We revised several sentences to avoid a grandiose tone. We emphasized several limitations of our method in discussion. We would appreciate any further suggestions for improvement from the reviewer.

We also think it is incorrect to say that innovation lies solely in the filtering step. Otherwise, the performance of PhaseNet-DAS will be similar to PhaseNet given the same filtering step. PhaseNet-DAS differs from single-channel phase pickers like PhaseNet, because it utilizes information from nearby channels to detect weak signals and prevent false picks. In the added results (Table S1, Figure 2, 4, S1-S4), we have compared the performance of PhaseNet and PhaseNet-DAS. PhaseNet-DAS's performance is significantly better than PhaseNet. Specifically, Figure S1 clearly shows that PhaseNet-DAS can pick phases on many more channels where PhaseNet can not.

METHOD:

Looking at this filtering step from some distance, I think it becomes clear that the rest of the proposed method, i.e., the training of a new network, has actually become redundant! After the filtering, i.e., the coherence-based cleaning of the single-trace picks, the problem of extracting seismic phases from the DAS data is actually already solved! The output of the filter is already what we need, and there is no obvious need to then build a new interpolation function (neural network) on top of that.

Hence, the whole purpose of half the method is actually not clear.

A: Because PhaseNet is not designed for DAS data and the performance of PhaseNet-DAS is significantly better than PhaseNet, so we did not include the comparison in our initial draft. We added new results in the revision to address the reviewer's concern that training of a new network based on the filtered labels are redundant. We can see improvements in both the first iteration training from PhaseNet to PhaseNet-DAS and the second iteration from PhaseNet-DAS v1 to PhaseNet-DAS v2 in association rate (Table 1), picked phases per event (Figure S1), and detected earthquakes (Figure 4, S3, S4). Given the same filtering using the phase associator GAMMA, the improvements demonstrate that the training is not redundant. Moreover, we can also see that the improvements from PhaseNet to PhaseNet-DAS are more significant in reducing false positive picks (higher associated rates), increasing phase picks per event (picking on more

channels), and detecting more events (sensitive to lower SNR events). This is because PhaseNet-DAS uses multi-channel information to pick P/S phase, which can reduce false picks on a few channels and increase the detection of coherent but weak signals across channels. Although PhaseNet is trained on many more seismic waveforms, it can not use the multi-channel information, thus has limited performance on DAS.

UNCERTAINTIES:

The authors' method also seems to produce earthquake locations. However, it seems like they are merely dots on a map. For many decades, earthquake locations have been reported with uncertainties, for obvious reasons.

Here, it is not clear to me where these uncertainties are and how the method could actually produce them. In case it can't, the method basically has no value for quantitative science. (It still makes impressive pictures.) Hence, this part needs some serious work.

A: We agree that location uncertainty is an important issue, but this is beyond the focus of this work. The method does not directly produce earthquake locations. The earthquake locations are the associated location based on the picked phase arrival times, which are similar to earthquake location based on phase arrivals. The earthquake location uncertainty depends on multiple factors, including phase picking uncertainty, velocity model uncertainty, cable geometries and azimuthal coverages, and optimization methods. We have analyzed the temporal accuracy of these automatic phase picks in Figure 3 using waveform cross-correlation. The other factors are independent of the phase picking model.

Because we only use one DAS array to constrain earthquake locations, the location uncertainty mainly comes from the limited azimuth coverage from the geometry of DAS. To provide more information about the location uncertainty, we added a grid-search result by plotting the whole loss surface using synthetic phases picks and velocity models in order to consider the effect of DAS geometry. From the loss surfaces in Figure S5, we can see that: (1) earthquake depths are hard to constrain using a single DAS array; (2) Events along the parallel line of the DAS array have a large location uncertainty (Figure S7a); (3) Events perpendicular to the DAS array have better constrained horizontal locations, though ambiguity may arise relative to which side of the DAS array (Figure S7c and d). The uncertainty also explains the location errors in Figure 5, especially the poor depth locations and the dispersion of events in the North-West corner.

I must admit that I may not have understood the method 100 %. However, in this case, I strongly suspect that a majority of the readers would not understand it either. Again, this goes hand in hand with the peculiar presentation style.

In summary, I do not think that there is sufficient innovation and creativity for a high-profile publication, and not enough technical information and detail for a more method-oriented journal.

A: We hope our point-by-point responses and revisions can address the raised concerns. We will release our code and model for testing the improvements of the retrained models. The model has already been used in several research projects in our group. We believe this can become beneficial to the community too.

Reviewer #3 (Remarks to the Author):

The paper presents a semi-supervised deep learning approach (PhaseNet-DAS) to pick seismic phase arrival times in distributed acoustic sensing (DAS) data. The study proposes the use of pre-trained PhaseNet to generate noisy pseudo-labels of P and S picks from individual DAS channels. Next, the study uses a phase association method (Gaussian Mixture Model Associator (GaMMA)) to distinguish noise and seismic arrivals from the pseudo-labels, essentially clustering respective phases into groups. Data augmentation (random flipping) is also added to increase the diversity of the dataset. By following this methodology, a large number of training datasets can be generated, and the quality of the dataset can be refined by repeating the procedure several times if desired. Consequently, the study proposes PhaseNet-DAS, which takes in two-dimensional (2D) DAS data, and predicts the relevant 2D P and S arrivals.

This study presents a major contribution to the DAS geophysics community. DAS is often clouded with inherent noise due to its highly sensitive and dense sensors. Existing machine learning (ML) pickers only work best on individual traces, which is likely to predict inconsistent phase picks when applied on 2D DAS data. As evident in Figure 1, PhaseNet-DAS seems to be able to predict the P and S picks with minimal noise. Furthermore, the semi-supervision nature of this study means that less manual work is needed to label 2D DAS signals. This contribution is significant because manually labeling 2D DAS signals can be highly laborious at times.

A: Thank you for recognizing the challenges in picking seismic phases on DAS and the innovation of our methods.

Comments/questions:

As DAS is not only used to monitor earthquakes, is PhaseNet-DAS limited to only detecting earthquakes? Can PhaseNet-DAS predict non-earthquake signals captured by DAS? Some examples of other seismic signals include vehicular traffic, mining blasts, thunderstorms, and microseismicity. I suggest discussing the limitations and/or feasibility of PhaseNet-DAS on detecting non-earthquake signals.

A: The current model is trained to detect only earthquake signals. This is because the PhaseNet model is trained to detect only earthquakes, the generated noisy labels are mostly earthquake signals, and we further applied phase association to filter out only earthquake signals. However, the semi-supervised learning approach could also be applied to other non-earthquake signals.

The semi-supervised learning approach is effective in utilizing large unlabeled DAS data. The limitation is that we need a pre-trained model on either seismic data or a small labeled DAS dataset for other non-earthquake signals. Here are some thoughts about developing deep learning models for non-earthquake signals the reviewer mentioned:

For vehicular traffic: since PhaseNet also picks many false picks on traffic signals, we could design some filtering criteria to select only picks from traffic, for example, setting the velocity model of phase association to common vehicle velocity. Then we can train a new PhaseNet-DAS model to pick the traffic signals.

For mining blasts: there are some manual labels on seismic data. We can retrain a PhaseNet model for detecting mining blasts, then follow the same semi-supervised approach to develop a PhaseNet-DAS model for mining blasts.

For thunderstorms: I am not familiar with thunderstorms signals on DAS. I am happy to discuss more about potential ways to develop models for detecting thunderstorms on DAS.

For microseismicity: There are several studies showing that PhaseNet works good on picking P and S phases of microseismicity. So we can directly apply PhaseNet and semi-supervised learning to train a PhaseNet-DAS model for microseismicity. I think the currently trained PhaseNet-DAS should also work for microseismicity, but maybe with reduced performance. We will release the model with the paper for testing on microseismicity.

For the deep learning training, how are the input 2D datasets (1024x1024) cut? What about cases when a cut dataset contains only S arrivals? Would this affect the training and prediction?

A: In training, we randomly sample a window from DAS data to generate many training samples. In inference, if the data can fit into GPU memory, we do not need to cut any windows. If the data can not fit into GPU memory, we will cut the whole DAS data into consecutive windows. The training includes samples containing only P or S arrivals, since we are randomly sampling the DAS data. This can make the model still work in prediction with only P or S phase, but performance will reduce due to limited context of waveforms information.

REVIEWER COMMENTS

Reviewer #1 (Remarks to the Author):

The paper “Seismic Arrival-time Picking on Distributed Acoustic Sensing Data using Semi-supervised Learning” by Zhu et al describes a deep learning method for seismic phase arrival time picking and event detection on DAS data. The revision is very comprehensive and addresses the critical comments I provided in my first review. I only have one larger point and a few minor comments remaining.

As far as I understand the experiments, all tests were performed on cataloged events, i.e., testing the sensitivity of the proposed models. However, for a full picture of the performance it would also be essential to quantify to which degree the models produce false positives. Without such an analysis, it is not clear whether the better detection performance only comes at the cost of a higher false positive rate. To clarify this, I’d suggest to add an analysis on continuous waveforms or a similar test addressing the question.

Upon reading the second review, I also want to clarify that in my opinion the contributions of this paper, in particular, the developed novel training scheme are far from trivial. In contrast to simple supervised learning, the proposed semi-supervised scheme is non-standard and the use of a single-station model together with a phase associator is an innovative approach. In addition, the work is of a high technical standard. While I agree with reviewer 2 regarding the language being exuberant in places, I think this revision of the manuscript has appropriately toned down the claims.

Given the comprehensive revisions to the manuscript, I recommend acceptance once the remaining points have been addressed.

Line numbers refer to the version with change highlighting.

- The manuscript states that as the model is fully convolutional, it can be applied to any input size. While technically correct, in practice U-Nets expose clear boundary artifacts that make applications to window sizes the model was not trained on not advisable. I’d suggest to add a comment to the manuscript whether such artifacts occur.
- L345: The statement that data quality and size matter more than hyperparameter tuning seems none-obvious to me. I’d suggest either providing a reference or removing the statement.
- Line 70: “million of pairs” → “million pairs”
- Lines 242 – 244: I’d recommend phrasing the sentence slightly more speculative as it’s not clear whether the same approach would work for the suggested tasks.
- Please add a table with the parameters chosen for GaMMA.
- Please specify the value of sigma used for the labels.
- Please clarify how overlaps on P and S waves are labeled, i.e., if $P_s + P_p > 1$.
- The link to the implementation currently (8th September) does not work.

We appreciate the constructive comments from the reviewer that helped us improve the manuscript. Based on the major comment about false positives, we have added a new test using one week's continuous data and updated the manuscript (Figures S6 and S7). Below, we provide a detailed point-by-point response.

Reviewer #1

The paper "Seismic Arrival-time Picking on Distributed Acoustic Sensing Data using Semi-supervised Learning" by Zhu et al describes a deep learning method for seismic phase arrival time picking and event detection on DAS data. The revision is very comprehensive and addresses the critical comments I provided in my first review. I only have one larger point and a few minor comments remaining.

As far as I understand the experiments, all tests were performed on cataloged events, i.e., testing the sensitivity of the proposed models. However, for a full picture of the performance it would also be essential to quantify to which degree the models produce false positives. Without such an analysis, it is not clear whether the better detection performance only comes at the cost of a higher false positive rate. To clarify this, I'd suggest to add an analysis on continuous waveforms or a similar test addressing the question.

A: Thank you for this suggestion. We have added a case using continuous data and updated the last paragraph of the section of earthquake monitoring (L188-L211). We note that due to the lack of ground truth for the number of earthquakes in real data, it is challenging to conduct a quantitative analysis. There are two indirect pieces of evidence showing the better detection performance does not come at the cost of a higher false positive rate.

First, the event data window (120s) contains a lot of noise. The association results in Table S1 gives an approximate estimation of false positive picks. The association ratio shows that the rate of false positive picks are not high (<10%), and PhaseNet-DAS has significantly fewer false positive picks compared to PhaseNet. Meanwhile, we also need to be cautious that the unassociated picks can be from true events (and, vice versa, the associated picks could also contain false positives), because GaMMA is not perfect especially when many events exist in one event sample, and we have to choose some thresholds, for example, a minimum of 500 picks per event.

Second, the rate of false positive events should be much lower after association filtering out false positive picks. From the example of the continuous data results. We can see well correlation with the routine seismic catalog in Figure S7. We added the discussion in the revision: "To assess the potential for false positive events, we

compared the associated earthquakes with events in routine earthquake catalogs. The histograms of temporal earthquake frequency in Figure S7 reveal a good correlation between events detected by the DAS array and seismic networks. Specifically, for events within ~ 0.5 degree of the DAS cable (Figure S7c), we can observe that earthquake frequencies vary from over 80 events to no events per six hours. Given the background noise generally does not change dramatically from day to day, this indicates that these new detections are less likely to be false detections from noise sources such as traffic.”

Evaluating false positives is a common challenge both for seismic data and DAS data, as deep learning models are generally more sensitive to weak signals and detect several times more events than the routine catalog. Comprehensively and qualitatively evaluating earthquake detection performance on real continuous data would require a well-designed benchmark dataset, which could be one direction of future research.

Upon reading the second review, I also want to clarify that in my opinion the contributions of this paper, in particular, the developed novel training scheme are far from trivial. In contrast to simple supervised learning, the proposed semi-supervised scheme is non-standard and the use of a single-station model together with a phase associator is an innovative approach. In addition, the work is of a high technical standard. While I agree with reviewer 2 regarding the language being exuberant in places, I think this revision of the manuscript has appropriately toned down the claims.

A: Thank you very much for recognizing the contribution of our work.

Given the comprehensive revisions to the manuscript, I recommend acceptance once the remaining points have been addressed.

Line numbers refer to the version with change highlighting.

- The manuscript states that as the model is fully convolutional, it can be applied to any input size. While technically correct, in practice U-Nets expose clear boundary artifacts that make applications to window sizes the model was not trained on not advisable. I'd suggest to add a comment to the manuscript whether such artifacts occur.

A: We agree that U-Nets could use boundary information to improve the prediction. For example, Figure 1 of Perk et al. (2023)'s work show that only two events near the two edges are picked. But this failed case is primary because we do not have closely space events (~ 5 s) in the training dataset. If we up-sample the waveform by a factor 2 or more, which is similar to doubling the event intervals, all events can be picked well.

On the other hand, using a long window size should prevent the model from to using the boundary information. Although the model may use the boundary information

to improve performance, we think a robust model should not deteriorate too much without using the edge information when using a long time window, since the edge information are not true information of waveforms. In the test of continuous data, we use a window of 200s (20,000 points) limited by GPU memory. The performance using a long window size looks good in Figure S6 and S7.

- L345: The statement that data quality and size matter more than hyperparameter tuning seems none-obvious to me. I'd suggest either providing a reference or removing the statement.

A: Thank you for the suggestion. That conclusion is based more on our empirical experience. We have removed this sentence.

- Line 70: "million of pairs" → "million pairs"

A: Corrected.

- Lines 242 – 244: I'd recommend phrasing the sentence slightly more speculative as it's not clear whether the same approach would work for the suggested tasks.

A: We have modified the sentence. "It would be promising to explore whether the semi-supervised approach could also serve in the development of deep learning models for other seismic signals on DAS data, including the detection of tremors and the picking of first motion polarities, where significant seismic archives are available"

- Please add a table with the parameters chosen for GaMMA.

A: We have added Table S2 of GaMMA parameters used.

- Please specify the value of sigma used for the labels.

A: We have added that: "We set sigma to 1.5 seconds in this work."

- Please clarify how overlaps on P and S waves are labeled, i.e., if $P_s + P_p > 1$.

A: We updated Eq 3: $P_n = \max(0, 1 - P_s - P_p)$, which is how we implement in the code. P_s and P_n stay the same.

- The link to the implementation currently (8th September) does not work.

A: We are cleaning the code and will soon make the repo public under the same link.

References:

Park, Y., Beroza, G. C., & Ellsworth, W. L. (2023). A mitigation strategy for the prediction inconsistency of neural phase pickers. *Seismological Society of America*, 94(3), 1603-1612.

REVIEWERS' COMMENTS

Reviewer #1 (Remarks to the Author):

The paper "Seismic Arrival-time Picking on Distributed Acoustic Sensing Data using Semi-supervised Learning" by Zhu et al describes a deep learning method for seismic phase arrival time picking and event detection on DAS data. As the first revision, the second revision comprehensively addresses the points raised in the review. The analysis of false positives is well-done and shows that indeed the sensitivity increases without substantial false detections. I want to point out that even the discussed false positive rate around 10 % is rather low in comparison to my own experience with broadband data.

Given the results, I now recommend the manuscript for publication in Nature Communication. This is a well-executed work and a well-written manuscript. It fills an important method gap, because automatic catalog generation from DAS data is currently not easily achievable with ready-made tools.

Response to REVIEWERS' COMMENTS

Reviewer #1 (Remarks to the Author):

The paper "Seismic Arrival-time Picking on Distributed Acoustic Sensing Data using Semi-supervised Learning" by Zhu et al describes a deep learning method for seismic phase arrival time picking and event detection on DAS data. As the first revision, the second revision comprehensively addresses the points raised in the review. The analysis of false positives is well-done and shows that indeed the sensitivity increases without substantial false detections. I want to point out that even the discussed false positive rate around 10 % is rather low in comparison to my own experience with broadband data.

Given the results, I now recommend the manuscript for publication in Nature Communication. This is a well-executed work and a well-written manuscript. It fills an important method gap, because automatic catalog generation from DAS data is currently not easily achievable with ready-made tools.

Response: Thank you for your positive comments. In terms of the false positive rate, it's important to note that this is based on an approximation using the association result as reference, because we do not have manual labels to calculate the true false positive rate. There are two factors that may affect the estimation of false positive rate. The first factor is the association algorithm is not perfect, which could associate some false picks as true picks or mis-associated true picks. We've set a minimum threshold of 500 picks to declare a true positive event in order to mitigate this issue. The second factor is PhaseNet-DAS already considers multi-channel information, which has a similar effect as association. This makes PhaseNet-DAS unlikely to predict false picks on just a few channels. For example, PhaseNet-DAS can effectively handle traffic noise which only exists within a small spatial range.